# ViewCo: Discovering Text-Supervised Segmentation Masks via Multi-View Semantic Consistency

**Pengzhen Ren[1], Changlin Li[2], Hang Xu[3], Yi Zhu[3], Guangrun Wang[4], Jianzhuang Liu[3], Xiaojun Chang[2], Xiaodan Liang[1,5]\***

[1]Sun Yat-sen University    [2]ReLER, AAII, University of Technology Sydney
[3]Huawei Noah's Ark Lab    [4]University of Oxford    [5]MBZUAI
pzhren@foxmail.com, {changlinli.ai,wanggrun,xdliang328}@gmail.com
{xu.hang,zhuyi36,liu.jianzhuang}@huawei.com, xiaojun.chang@uts.edu.au

## Abstract

Recently, great success has been made in learning visual representations from text supervision, facilitating the emergence of text-supervised semantic segmentation. However, existing works focus on pixel grouping and cross-modal semantic alignment, while ignoring the correspondence among multiple augmented views of the same image. To overcome such limitation, we propose multi-**View Co**nsistent learning (ViewCo) for text-supervised semantic segmentation. Specifically, we first propose text-to-views consistency modeling to learn correspondence for multiple views of the same input image. Additionally, we propose cross-view segmentation consistency modeling to address the ambiguity issue of text supervision by contrasting the segment features of Siamese visual encoders. The text-to-views consistency benefits the dense assignment of the visual features by encouraging different crops to align with the same text, while the cross-view segmentation consistency modeling provides additional self-supervision, overcoming the limitation of ambiguous text supervision for segmentation masks. Trained with large-scale image-text data, our model can directly segment objects of arbitrary categories in a zero-shot manner. Extensive experiments show that ViewCo outperforms state-of-the-art methods on average by up to 2.9%, 1.6%, and 2.4% mIoU on PASCAL VOC2012, PASCAL Context, and COCO, respectively. [1]

## 1 Introduction

Recently, vision-language contrastive learning (Radford et al. (2021); Li et al. (2021a)) has attracted a lot of attention because it can obtain more generalized feature representation. And at the same time, it can also make use of abundant image-text pairs to avoid labor-intensive annotation costs. Vision-language pre-training (VLP) models have exhibited impressive potential in various visual (Xu et al. (2022); Mu et al. (2021); Radford et al. (2021)) and multimodal (Wang et al. (2021b); Kim et al. (2021)) tasks, including text-supervised semantic segmentation (Xu et al. (2022); Ghiasi et al. (2021); Xu et al. (2021); Zabari & Hoshen (2021); Zhou et al. (2021a), which uses text instead of traditional dense labels for supervision to achieve zero-shot semantic segmentation. It provides a feasible solution for learning segmentation masks without mask annotation.

However, existing works with CLIP-based (Radford et al. (2021)) segmentation (Xu et al. (2022; 2021); Zhou et al. (2021a)) mainly focus on pixel grouping or cross-modal semantic alignment. They have the following two obvious limitations: *(i)* the excessive strictness of image-text correspondence; and *(ii)* the ambiguity of text description. *First*, in vanilla vision-language contrastive learning, each image-text

All Hallows Bronx Parks Advocacy Day, all Halloween in parks and squares.

Figure 1: Illustration of text description ambiguity. Text descriptions are highly abstract and difficult to be semantically aligned with images. Cross-view semantic consistency modeling can effectively alleviate the effect of the text description ambiguity issue.

---

*Corresponding author.
[1]Code release: https://github.com/pzhren/ViewCo

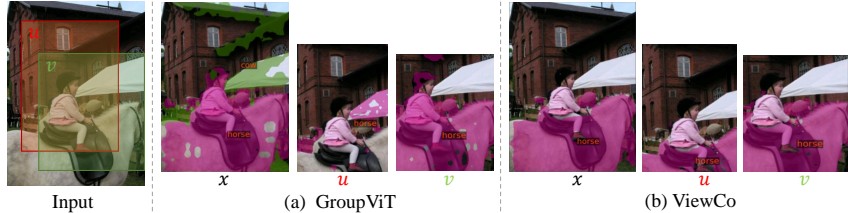

Figure 2: The consistent comparison of semantic segmentation results in multiple views of a "horse". (a) GroupViT: the semantic segmentations of different views are inconsistent. (b) ViewCo: the semantic segmentations of different views are much more consistent. Here, $x$, $u$, and $v$ represent the segmentation results on the original image and views $u$, $v$, respectively.

pair is regarded as a unique positive pair, while all the other combinations are regarded as negative ones. This image-text correspondence is actually too rigorous. In fact, one textual description may correspond to different images. The excessive strictness is not conducive to the model learning high-level cross-modal semantic correspondences. Therefore, more relaxed vision-language contrastive learning needs to be considered. *Second*, the ambiguity of textual descriptions is also a key challenge. Compared with the traditional semantic segmentation pipeline that uses dense annotations as supervision information (Touvron et al. (2021); Ren et al. (2022)), the CLIP-based segmentation methods (Xu et al. (2022; 2021); Zhou et al. (2021a)) use text as supervision, which is easier to access but more noisy and ambiguous.

This is mainly because compared with traditional segmentation annotations, text descriptions are often more abstract and do not contain location information. Moreover, the background in the image is usually ignored in the description. In some cases, the objects in the image do not even exist in the text description (see Figure 1). Such ambiguity is common in the textual supervision in vision-language pre-training. In the semantic segmentation task, the ambiguity of textual supervision makes the segmented object-label correspondence very fragile. Therefore, Fully mining the information carried by the dataset itself may need to be considered.

On the other hand, visual self-supervision (Caron et al. (2021); He et al. (2022); Chen et al. (2020a); Zhou et al. (2021b)) has been widely used for visual pre-training. It includes two categories: reconstructing masked images (He et al. (2022); Zhou et al. (2021b)) and multicrop image contrast (Caron et al. (2021); Chen et al. (2020a)). For example, SLIP (Mu et al. (2021)) introduces contrastive learning of multicrop visual consistency for VLP. MaskCLIP (Dong et al. (2022)) introduces a visual self-supervised task of reconstructing masked images. They utilize visual self-supervision to provide more useful information for VLP models. However, the semantic consistency of multiple views of an image in segmentation and cross-modal contrast have not received enough attention and research.

Based on the above observations, in this paper, we explore the impact of multi-view semantic consistency on the task of text-supervised semantic segmentation through visual self-supervision. To this end, we propose multi-**View Co**nsistency learning (ViewCo), which aims at discovering text-supervised segmentation masks via multi-view semantic consistency. Specifically, we propose *text-to-views consistency modeling* to alleviate the excessive strictness of image-text correspondence in vanilla vision-language contrastive learning. It enables the model to benefit from the dense assignment of visual features by encouraging different crops to align with the same text. This relaxed one-to-many contrast mechanism also facilitates the learning of multi-view consistent semantics, enabling the model to acquire high-level cross-modal alignment capabilities. Moreover, as shown in Figure 1, to alleviate the ambiguity issue of textual supervision, we propose *cross-view segmentation consistency modeling*. It overcomes the limitation imposed by textual ambiguity by providing additional self-supervision to vision-language contrastive learning via cross-view segmentation consistency. ViewCo uses the proposed text-to-views consistency modeling for vision-language cross-modal contrastive learning and additionally enables cross-view segmentation consistency modeling by contrasting the segment features of Siamese visual encoders. As shown in Figure 2, with the help of the two consistency modeling schemes, ViewCo establishes a solid semantic correspondence in different views, and the semantics in different views maintain a good consistency. The semantic consistency of GroupViT in different views is difficult to guarantee.

Overall, ViewCo's design is simple and effective. We train it on large-scale image-text pair datasets CC12M (Changpinyo et al. (2021)) and YFCC (Thomee et al. (2016)). In the inference stage, we

use the similarity scores between the segmentation embeddings generated by the teacher network and the label prompts to assign labels to the image masks for zero-shot semantic segmentation. Compared with the state-of-the-art methods, ViewCo achieves an average improvement of 2.9%, 1.6%, and 2.4% mIoU on PASCAL VOC2012, PASCAL Context, and COCO, respectively. Our contributions can be summarized as follows:

- We propose a novel one-to-many text-to-views consistency modeling that improves the model's ability of high-level cross-modal semantic alignment by encouraging different crops of an image to align with the same text.
- To alleviate the problem of supervision failure that may arise from text ambiguity, we propose cross-view segmentation consistency modeling to provide additional self-supervision for the vision branch and encourage the model to generate consistent segmentation masks for different views.
- ViewCo consistently outperforms the state-of-the-art methods on PASCAL VOC2012, PASCAL Context, and MS-COCO when pre-trained on CC12M or CC12M+YFCC.

## 2 RELATED WORK

**Vision-Language Pretraining.** In recent years, vision-language pre-training models (Chen et al. (2020b); Desai & Johnson (2021); Li et al. (2020a; 2021a; 2020b)) have developed rapidly with the help of large-scale image-text pair data available on the Internet. Recently, VLP models such as CLIP (Radford et al. (2021)), ALIGN (Li et al. (2021a)), and SLIP (Mu et al. (2021)) have made great progress in visual representation learning by using contrastive learning. And they have been successfully transferred to various downstream tasks, such as visual question answering (Antol et al. (2015); Zhou et al. (2020)) and visual reasoning (Zellers et al. (2019)). In particular, CLIP (Radford et al. (2021)) uses the image-text matching relationship for contrastive learning, and the learned model can be directly transferred to ImageNet classification (Deng et al. (2009)) in a zero-shot manner without any fine-tuning. This success is also found on zero-shot semantic segmentation (Xu et al. (2022)). However, the one-to-one contrastive learning mechanism between image and text in the vanilla VLP pipeline is too strict, which is not conducive to the model learning high-level cross-modal semantic alignment. Based on the above observations, this paper proposes one-to-many text-to-views consistency modeling. It relaxes the original one-to-one correspondence by encouraging different crops of an image to match the same text, allowing the model to benefit from the dense assignment of the visual features.

**Visual Self-Supervision.** This framework relies on the information carried by the image itself for self-supervision without any additional annotation information. Visual self-supervision is mainly divided into generative (He et al. (2022); Bao et al. (2021)) and contrastive (He et al. (2020a); Caron et al. (2021); Chen et al. (2020a)). A generative model allows the model to learn the feature representation of the image by reconstructing the masked image. Contrastive models focus more on learning-centric global representations. Since semantic segmentation requires dense prediction of images, generative models may not help much because they destroy the original structure and information of images. On the other hand, the contrastive visual self-supervised model can provide the required multi-view features for ViewCo's text-to-views consistency modeling. Moreover, this visual contrastive learning can provide additional visual self-supervision information for the VLP model to alleviate the risk of supervision failure caused by text ambiguity. Therefore, this paper focuses on the help of visual contrastive learning for semantic segmentation consistency.

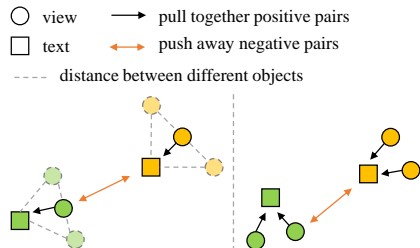

**Consistent Semantics.** Capturing consistent semantics is one of the main challenges shared by many tasks such as cross-modality and visual understanding. Vision-language contrastive learning (Radford et al. (2021)) is essential to encode different modal data into the same feature space and enforces the features sharing the same semantics to get closer, and the features with different semantics to be pushed away. Similarly, multicrop image-level semantic consistency is

Figure 3: Comparison of single-view text-to-image contrastive learning (left) and multi-view text-to-views contrastive learning (right).

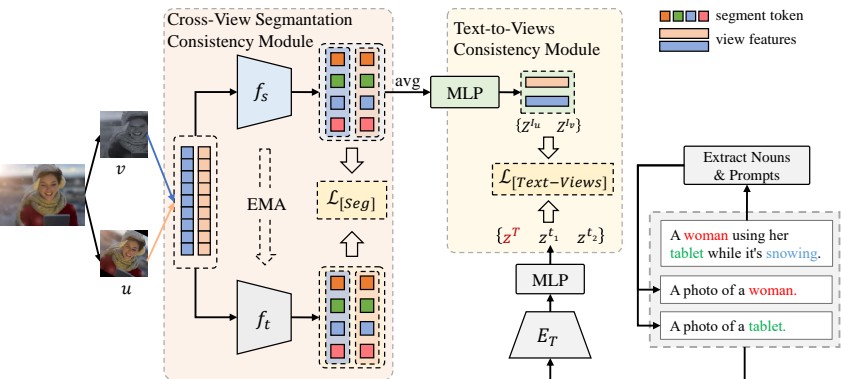

Figure 4: Framework of ViewCo. It is mainly composed of a cross-view segmentation consistency module and a text-to-views consistency module. The visual branch adopts a visual self-supervised model, which consists of teacher $f_t$ and student $f_s$ networks with the same structure. $f_t$ and $f_s$ are the bottom-up segmentation backbone that outputs segment features of the image.

also the core idea of visual self-supervised contrastive learning (Caron et al. (2021); Chen et al. (2020a); He et al. (2020a)). For example, DenseCL (Wang et al. (2021a)) performs pixel-level dense contrastive learning on dense output vectors from multiple views, which is not helpful to the learning of high-level global semantic information. Further, GroupViT (Xu et al. (2022)) uses text as supervision and achieves pixel grouping by capturing the contextual consistent semantics of images. However, in the text-supervised semantic segmentation task, the ambiguous properties of text relative to dense annotations result in that the semantic consistency of images sharing the same semantics cannot be sufficiently guaranteed in the embedding space. Furthermore, the strict one-to-one correspondence between image and text in the vanilla VLP model is also not conducive to the true alignment of high-level cross-modal semantics. Figure 3 (left) illustrates the above observation: although one of the views of an image (*e.g.*, the solid circle) is already close to the corresponding text embedding, other views (*e.g.*, the dashed circles) may still be far away. Previous VLP methods generally only focus on the alignment of a single view with text. In contrast, as shown in Figure 3 (right), ViewCo focuses on text-to-views consistency modeling, doing one-to-many matching in cross-modal contrastive learning.

## 3 MULTI-VIEW CONSISTENT LEARNING

As shown in Figure 4, our ViewCo is mainly composed of a cross-view segmentation consistency module and a text-to-views consistency module. We describe these two modules in Sections 3.1 and 3.2, respectively, and summarize the final loss function in Section 3.3.

### 3.1 CROSS-VIEW SEGMENTATION CONSISTENCY MODULE

As shown in Figure 4 (left), given a batch of image-text pairs $\{(x_i^I, x_i^T)\}_{i=1}^B$, two random augmentations are performed on the input image $x_i^I$, generating two warped views $u$ and $v$. We use GroupViT (Xu et al. (2022)) as the bottom-up segmentation backbone of ViewCo, where each view is segmented into $K$ segment tokens. For each of the views (e.g., $u$), this process is expressed as: $Z_{\text{Seg}}^{u_s} = \{Z_{\text{seg}_k}^{u_s}, k = 1, ..., K\} = f_s(u) \in \mathbb{R}^{K \times d}$, where $Z_{\text{seg}_k}^{u_s} \in \mathbb{R}^d$ is the $k$-th segment feature from $f_s$, and $d$ is the dimensionality of the segment feature. Similarly, we have $Z_{\text{seg}_k}^{v_s}$ and the segment features $Z_{\text{seg}_k}^{u_t}$ and $Z_{\text{seg}_k}^{v_t}$ from the teacher network $f_t$. We update the parameters of $f_t$ using the exponential moving average (EMA) He et al. (2020b) of the parameters of $f_s$. For example, let $\theta_i$ and $\bar{\theta}_i$ be the parameters of $f_s$ and $f_t$ at training step $i$, respectively, and then $\bar{\theta}_i$ is updated as: $\bar{\theta}_i = \alpha \bar{\theta}_{i-1} + (1 - \alpha)\theta_i$, where $\alpha$ is a hyper-parameter for smoothing the update. In addition, the standard contrastive loss function, called InfoNCE (Oord et al. (2018)), is considered in this paper, for an encoded query $q$ and a set of encoded samples $k = \{k_0, k_1, k_2, ...\}^N$ that are the keys of a dictionary, we have:

$$\mathcal{L}_{NCE}(q, k) = -\log \frac{\exp(q \cdot k_+/\tau)}{\sum_{i=0}^N \exp(q \cdot k_i/\tau)}, \tag{1}$$

where $\tau$ is a learnable temperature parameter. And $q$ and $k_+$ are positive pairs, and the other $(N-1)$ pairs are negative.

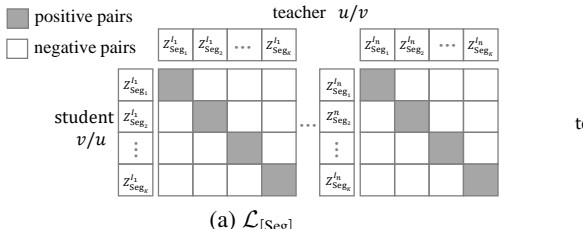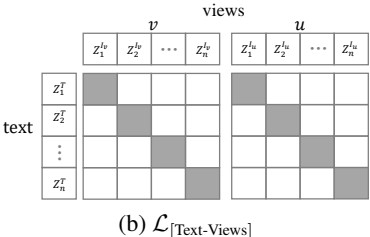

$$(a)\ \mathcal{L}_{[\text{Seg}]} \qquad\qquad\qquad (b)\ \mathcal{L}_{[\text{Text-Views}]}$$

Figure 5: Illustration of the contrastive loss of (a) cross-view segmentation consistency modeling and (b) text-to-views consistency modeling. $Z_{\text{Seg}_k}^{I_i}$ is the $k$-th semantic feature of the $i$-th image (*i.e.*, view $u$ or $v$). $Z_i^{I_v}$ and $Z_i^{I_u}$ are the embeddings of the views $v$ and $u$ of the $i$-th image, respectively.

Intuitively, the segment features obtained from different crops of the same image should be roughly the same, *i.e.*, cross-view segmentation consistency. To this end, for the semantic segmentation task, we replace the image-level contrastive learning in previous methods (Caron et al. (2021); Zhou et al. (2021b)) with cross-view segmentation consistency learning within images. Therefore, we define the minimization training objective of the cross-view segmentation consistency module in ViewCo as:

$$\mathcal{L}_{[\text{Seg}]}^{t\leftrightarrow s} = \mathcal{L}_{[\text{Seg}]}^{t\to s} + \mathcal{L}_{[\text{Seg}]}^{s\to t}. \tag{2}$$

It is a bi-directional contrast loss between the segment features from the teacher $f_t$ and the student $f_s$. $\mathcal{L}_{[\text{Seg}]}^{t\to s}$ considers two pairs of views (*i.e.*, $(u_t, v_s)$ and $(v_t, u_s)$) outputted by $f_t$ and $f_s$. The segment features of $(u_t, v_s)$ from the same image are multiplied ($Z_{\text{seg}}^{u_t} \cdot Z_{\text{seg}}^{v_s\ T} \in \mathbb{R}^{K\times K}$) after $l_2$ normalization. In the image branch of ViewCo, we use the EMA policy for parameter updates, so the learnable grouping tokens on the corresponding position IDs of different views of the same image are highly correlated, and they have the same semantics. Therefore, the semantic pairs $\{(Z_{\text{seg}_i}^{u_t}, Z_{\text{seg}_i}^{v_s}), i = 1, ..., K\}$ on the diagonal are regarded as positive, and the other $K(K-1)$ pairs $\{(Z_{\text{seg}_i}^{u_t}, Z_{\text{seg}_j}^{v_s}), i, j = 1, ..., K, i \neq j\}$ are regarded as negative. Therefore, the contrastive loss $\mathcal{L}_{[\text{Seg}]}^{t\to s}$ of the teacher-to-student segment features is defined as $\mathcal{L}_{[\text{Seg}]}^{t\to s} = \mathcal{L}_{u_t\to v_s} + \mathcal{L}_{v_t\to u_s}$, more specifically:

$$\mathcal{L}_{[\text{Seg}]}^{t\to s} = -\frac{1}{KB}\sum_{i=1}^{B}\sum_{k=1}^{K}(\mathcal{L}_{NCE}(Z_{\text{seg}_k}^{u_t}, \{Z_{\text{seg}_k}^{v_s}\}_{k=1}^{K}) + \mathcal{L}_{NCE}(Z_{\text{seg}_k}^{v_t}, \{Z_{\text{seg}_k}^{u_s}\}_{k=1}^{K})). \tag{3}$$

Similarly, the contrastive loss $\mathcal{L}_{[\text{Seg}]}^{s\to t}$ of the student-to-teacher segment features is defined as $\mathcal{L}_{[\text{Seg}]}^{s\to t} = \mathcal{L}_{u_s\to v_t} + \mathcal{L}_{v_s\to u_t}$, more specifically:

$$\mathcal{L}_{[\text{Seg}]}^{s\to t} = -\frac{1}{KB}\sum_{i=1}^{B}\sum_{k=1}^{K}(\mathcal{L}_{NCE}(Z_{\text{seg}_k}^{u_s}, \{Z_{\text{seg}_k}^{v_t}\}_{k=1}^{K}) + \mathcal{L}_{NCE}(Z_{\text{seg}_k}^{v_s}, \{Z_{\text{seg}_k}^{u_t}\}_{k=1}^{K})). \tag{4}$$

Figure 5a shows the positive and negative pairs for cross-view segmentation consistency learning in the vision branch.

### 3.2 TEXT-TO-VIEWS CONSISTENCY MODULE

Previous methods (Radford et al. (2021); Xu et al. (2022)) build visual-linguistic semantic correspondences by performing a contrastive loss on image-text pairs. In this paper, we consider the contrastive learning between multiple views and text, using one-to-many text-to-views consistency modeling instead of one-to-one text-to-image contrastive learning. The model learns to capture intra-modal and inter-modal semantic consistency through the alignment of multi-view images and text.

Specifically, for a given image-text pair $(x_i^I, x_i^T)$, by performing two different augmentations to the input image, we have a triplet $(u_i, v_i, x_i^T)$. As shown in Figure 4 (right), in the training phase, we take the output $(Z_i^u, Z_i^v)$ of the view pair $(u_i, v_i)$ through the student network $f_s$ and the output $Z_i^T$ of the text encoder $E_T$ to calculate the contrastive loss respectively. The visual embeddings $(Z_i^u, Z_i^v)$ and text embedding $Z_i^T$ are mapped to the same feature space through two MLPs, respectively, before performing the final $l_2$ regularization. This procedure is represented as:

$Z_i^{I_u} = \text{MLP}(\text{AvgPool}(Z_{[\text{Seg}]}^{u_i})), Z_{[\text{Seg}]}^{u_i} = f_s(u_i); Z_i^{I_v} = \text{MLP}(\text{AvgPool}(Z_{[\text{Seg}]}^{v_i})), Z_{[\text{Seg}]}^{v_i} = f_s(v_i).$
The multi-view feature $Z_i^I = \{Z_i^{I_u}, Z_i^{I_v}\}$ and text embedding $Z_i^T$ constitute positive pairs, and the other $2B(B-1)$ pairs are negative pairs. The contrastive loss of text-to-views consistency modeling is defined as follows:

$$\mathcal{L}_{I_{\{u,v\}} \leftrightarrow T} = \mathcal{L}_{I_{\{u,v\}} \to T} + \mathcal{L}_{T \to I_{\{u,v\}}}, \tag{5}$$

where the contrastive loss of views $I_{\{u,v\}}$-to-text is defined as:

$$\mathcal{L}_{I_{\{u,v\}} \to T} = -\frac{1}{KB} \sum_{i=1}^{B} \sum_{k=1}^{K} (\mathcal{L}_{NCE}(Z_i^{I_u}, \{Z_i^T\}_{i=1}^B) + \mathcal{L}_{NCE}(Z_i^{I_v}, \{Z_i^T\}_{i=1}^B)). \tag{6}$$

and the contrastive loss of text-to-views $I_{\{u,v\}}$ is defined as:

$$\mathcal{L}_{T \to I_{\{u,v\}}} = -\frac{1}{KB} \sum_{i=1}^{B} \sum_{k=1}^{K} (\mathcal{L}_{NCE}(Z_i^T, \{Z_i^{I_u}\}_{i=1}^B) + \mathcal{L}_{NCE}(Z_i^T, \{Z_i^{I_v}\}_{i=1}^B)). \tag{7}$$

Additionally, in order to further enhance the association between multi-view semantics and text semantics, we also compute the multi-label image-text contrastive loss (Xu et al. (2022)) of multi-view and "prompted text" pairs $\{(Z_i^{I_u}, \{Z_i^{t_m}\}_{m=1}^M)_{i=1}^B, (Z_i^{I_v}, \{Z_i^{t_m}\}_{m=1}^M)_{i=1}^B\}$, where $\{Z_i^{t_m}\}_{m=1}^M$ are the embeddings of the additional $M$ text prompts $\{T_i^m\}_{m=1}^M$ generated by the $i$-th text $x_i^T$ according to the "prompt engineering" mechanism (Radford et al. (2021)). $(Z_i^{I_u}, \{Z_i^{t_m}\}_{m=1}^M)$, $i.e.$, the embedding of the $i$-th image view $u$ and the generated $M$ text embeddings $\{Z_i^{t_m}\}_{m=1}^M$ are positive pairs, and the other combinations are negative pairs. Therefore, similar to Eq.(5), the multi-label contrastive loss of multi-view $I_{\{u,v\}}$ and multi-prompt $\{T^m\}_{m=1}^M$ is defined as:

$$\mathcal{L}_{I_{\{u,v\}} \leftrightarrow \{T^m\}_{m=1}^M} = \mathcal{L}_{I_{\{u,v\}} \to \{T^m\}_{m=1}^M} + \mathcal{L}_{\{T^m\}_{m=1}^M \to I_{\{u,v\}}}. \tag{8}$$

First, the views-to-prompts loss is the average of the losses of the two views. Considering a single view, e.g. $u$, the contrastive loss of $u$ to all the prompts is defined as:

$$\mathcal{L}_{I_u \to \{T^m\}_{m=1}^M} = -\frac{1}{B} \sum_{i=1}^{B} \left( \log \frac{\sum_{m=1}^M \exp(Z_i^{I_u} \cdot Z_i^{t_m}/\tau)}{\sum_{m=1}^M \sum_{j=1}^B \exp(Z_i^{I_u} \cdot Z_j^{t_m}/\tau)} \right). \tag{9}$$

Second, the contrastive loss of multi-prompt-to-views is defined as:

$$\mathcal{L}_{\{T^m\}_{m=1}^M \to I_{\{u,v\}}} = -\frac{1}{2MB} \sum_{m=1}^{M} \sum_{i=1}^{B} (\mathcal{L}_{NCE}(Z_i^{t_m}, \{Z_i^{I_u}\}_{i=1}^B) + \mathcal{L}_{NCE}(Z_i^{t_m}, \{Z_i^{I_v}\}_{i=1}^B)). \tag{10}$$

In particular, a similar work to our text-to-views consistency module is DeCLIP (Li et al. (2021b)). It believes that the text description may only be a small part of the image, so in addition to the global view in CLIP (Radford et al. (2021)), DeCLIP also adds a local view for image self-supervision, which may cause information leakage. In addition, DeCLIP uses EDA (Wei & Zou (2019)) as a text augmentation strategy. The augmented text still contains multiple semantics, which is not helpful to the alignment of local semantics in segmentation tasks. In contrast, ViewCo uses self-supervision of two local views to ensure the difficulty of the task, while using a "prompt engineering" mechanism to obtain an augmented text with a single semantic. Combining one-to-many alignment can help ViewCo to better mine consistent segmentation semantics in images.

### 3.3 Overall Loss Function

Finally, the total loss of ViewCo is the sum of the cross-view segmentation consistency contrastive loss and the two cross-modal contrastive losses:

$$\mathcal{L} = \mathcal{L}_{[\text{Seg}]}^{t \leftrightarrow s} + \mathcal{L}_{I_{\{u,v\}} \leftrightarrow T} + \mathcal{L}_{I_{\{u,v\}} \leftrightarrow \{T^m\}_{m=1}^M}. \tag{11}$$

## 4 Experiments

### 4.1 Implementation Details

**Architecture.** In the cross-view segmentation consistency module, $f_t$ and $f_s$ have the same network structure. The parameters of $f_t$ are updated using the exponential moving average of the parameters

| Arch | Pre-training | | | Zero-Shot | Transfer (mIoU (%)) | | |
| | Model | Dataset | Supervision | | PASCAL VOC | PASCAL Context | COCO |
|---|---|---|---|---|---|---|---|
| ViT | DeiT | ImageNet | class | ✗ | 53.0 | 35.9 | - |
| | DINO | ImageNet | self | ✗ | 39.1 | 20.4 | - |
| | DINO | CC12M+YFCC | self | ✗ | 37.6 | 22.8 | - |
| | MoCo | ImageNet | self | ✗ | 34.3 | 21.3 | - |
| | MoCo | CC12M+YFCC | self | ✗ | 36.1 | 23.0 | - |
| CLIP | CLIP | LAION-20M | text | ✓ | - | 13.5 | 8.2 |
| | GroupViT | CC12M | text | ✓ | 41.1 | 18.2 | 18.4 |
| | GroupViT[1] | CC12M+YFCC | text | ✓ | 51.2 | 22.3 | 20.9 |
| | SLIP | LAION-20M | text & self | ✓ | - | 12.3 | 8.8 |
| | CLIP-MAE | LAION-20M | text & self | ✓ | - | 16.8 | 11.8 |
| | MaskCLIP | LAION-20M | text & self | ✓ | - | 17.7 | 11.8 |
| | ViewCo (ours) | CC12M | text & self | ✓ | **45.7** | **20.8** | **20.6** |
| | ViewCo (ours) | CC12M+YFCC | text & self | ✓ | **52.4** | **23.0** | **23.5** |

Table 2: Comparisons with recent methods. Zero-shot means that the model is directly transferred to the semantic segmentation task without any fine-tuning on the target dataset.

of $f_s$. We use GroupViT (Xu et al. (2022)) with two stages as the backbone for semantic feature extraction of ViewCo's visual branch. It is built on ViT-S (Dosovitskiy et al. (2020); Touvron et al. (2021)) with 12 Transformer layers. The input image size is $224 \times 224$, the patch size is $16 \times 16$, and the hidden dimensionality is 384. The 2-stage GroupViT finally outputs 8 segment tokens, *i.e.,* $K = 8$). Following Radford et al. (2021), ViewCo's text encoder $E_T$ consists of 12 Transformer layers with a hidden feature dimensionality of 256.

**Training and Inference.** In the training phase, we use CC12M (Changpinyo et al. (2021)) and the filtered YFCC (Thomee et al. (2016)) as training datasets, which contain 12M and 14M image-text pairs, respectively. See A.1 of the supplementary material for more training details. In the inference phase, following (Xu et al. (2022); Radford et al. (2021)), the image is segmented by associating the image patches with the $K$ segment tokens outputted by the teacher network $f_t$. The semantics in the images are further classified by computing the similarity of the $K$ visual-semantic embeddings to the text embeddings generated from the class labels of the test dataset.

**Zero-Shot Transfer to Semantic Segmentation.** We evaluate ViewCo on the task of zero-shot transfer to semantic segmentation on the validation sets of PASCAL VOC 2012 (Everingham et al. (2010)), PASCAL Context (Mottaghi et al. (2014)) and COCO Stuff (Lin et al. (2014)) datasets. The three datasets contain 20, 59, and 80 foreground classes and an additional background class, respectively. During inference, following GroupViT (Xu et al. (2022)), ViewCo predicts only the foreground classes by thresholding the softmax-normalized-similarity between the embedding of the outputted image segments and the text segmentation labels. The thresholds on PASCAL VOC 2012, PASCAL Context, and COCO are set to 0.95, 0.35, and 0.95, respectively. We resize each input image to have a shorter side of 448.

## 4.2 COMPARISONS WITH RECENT METHODS

We first compare the performance of ViewCo with some ViT-S-based zero-shot baselines. Then, to further evaluate the performance of ViewCo on the zero-shot semantic segmentation task, we compare ViewCo with some fully supervised transfer and CLIP-based models.

**Comparison with Zero-Shot Baselines.** Table 1 shows the performance comparison of ViewCo and zero-shot baselines on PASCAL VOC 2012. Among them, the four ViT-based baselines train vision and text encoders through the image-text contrastive loss defined in CLIP (Radford et al. (2021)). They adopt four different pixel grouping methods: pixel-wise, K-means, Mean-shift (Comaniciu & Meer (2002)), and Spectral clustering (Shi & Malik (1997)) respectively. And GroupViT (Xu et al. (2022)) uses the bottom-up patch grouping mecha-

| Arch. | Method | Mask mIoU (%) |
|---|---|---|
| ViT | pixel-wise | 20.1 |
| ViT | K-means | 25.0 |
| ViT | Mean-shift | 20.7 |
| ViT | Spectral clustering | 19.7 |
| GroupViT | - | 51.2 |
| ViewCo (ours) | - | **52.4** |

Table 1: Comparison with zero-shot baselines on PASCAL VOC 2012.

nism. As shown in Table 1, ViewCo significantly outperforms the CLIP-trained ViT and GroupViT (52.4% *vs.* 51.2%) baselines. It is worth noting that ViewCo and GroupViT adopt the same segmentation backbone, indicating that ViewCo can effectively improve the model's ability of segmentation and cross-modal semantic alignment with the help of the two consistent semantic modelings.

**Comparison with Other SoTA Methods.** These methods include one fully supervised baseline DeiT (Touvron et al. (2021)), two visual self-supervised baselines DINO (Caron et al. (2021)) and MoCo (He et al. (2020a)), two vision-language contrastive learning baselines CLIP (Radford et al. (2021)) and GroupViT (Xu et al. (2022)), two vision-language contrast and visual self-supervised learning combined baselines SLIP (Mu et al. (2021)) and CLIP-MAE (Dong et al. (2022)), and one vision-language contrast and self-distillation combined baseline MaskCLIP (Dong et al. (2022)).

Table 2 shows the mIoU performance comparison between ViewCo and the SoTA methods on PAS-CAL VOC 2012, PASCAL Context, and COCO validation sets. ViewCo significantly outperforms them on all three datasets. Compared to GroupViT, when pre-trained on CC12M, ViewCo achieves a 4.6% mIoU improvement on PASCAL VOC. Similarly, when pre-trained on CC12M+YFCC, ViewCo achieves a 2.6% mIoU improvement on COCO compared to GroupViT. Similar to ViewCo, SLIP, MaskCLIP, and CLIP-MAE all use additional supervision information in the vision branch of the VLP models. Compared with them, ViewCo still has clear advantages in PASCAL Context and COCO. In addition, ViewCo obtains segmentation performance close to the fully supervised DeiT on PASCAL VOC, which again demonstrates the effectiveness of ViewCo for zero-shot semantic segmentation.

### 4.3 ANALYSIS

In this section, for the convenience of comparison, we use CC12M as the pre-training dataset by default for the ablation of ViewCo components, qualitative analysis, and image classification performance comparison. GroupViT (Xu et al. (2022)) is used as the baseline for ViewCo.

**Image-Level Contrast *vs*. Semantic-Level Contrast.** To ablate the role of the cross-view segmentation consistency module in the vision branch, we add an image-level contrastive module to GroupViT in the visual branch, where we first calculate the average of the $K$ segment tokens outputted by the teacher and student networks, and then perform contrastive learning. For ViewCo, we remove the text-to-views consistency module and directly average pool the multi-view features outputted by the student network. To be consistent with GroupViT, we use the pooled visual features for contrastive learning with text embeddings. As shown in Table 3, adding a visual self-supervised module for vision-language contrastive

|  | Visual branch | COCO mIoU (%) |
|---|---|---|
| GroupViT | - | 18.4 |
| GroupViT+ | image-level | 18.6 |
| ViewCo | semantic-level | **19.1**(0.7↑) |

Table 3: Image-level contrast *vs.* semantic-level contrast. "-" indicates that no visual self-supervision module is used. GroupViT+ represents modifying the corresponding component in GroupViT.

learning can improve the performance of the model on semantic segmentation by improving the quality of visual feature learning. Furthermore, the improved performance (*i.e.*, 19.1 *vs.* 18.6) of semantic-level learning relative to image-level contrastive learning suggests that the cross-view segmentation consistency module can further improve the performance by capturing the consistency of cross-view semantic segmentation.

**Vision-Language Contrast: Text-to-Image *vs*. Text-to-Views.** We further ablate the text-to-views consistency module in ViewCo. In single-view vision-language contrastive learning, we use the average embedding of multi-view features outputted by the student network and the text embedding for contrastive learning during training. As shown in Table 4, text-to-views consistency modeling significantly improves the performances of the models compared to single-view text-to-image (*i.e.*, 1.1% and 1.5%). This indicates that text-to-views consistency modeling has better high-level semantic alignment capabilities than text-to-image single-view modeling. This is exactly what previous methods of single-view vision-language contrastive learning do not have.

|  | Visual branch | Vision-language contrast | COCO mIoU (%) |
|---|---|---|---|
| GroupViT+ | image-level | single | 18.6 |
| GroupViT+ | image-level | multiple | **19.7** (1.1↑) |
| ViewCo | semantic-level | single | 19.1 |
| ViewCo | semantic-level | multiple | **20.6** (1.5↑) |

Table 4: Vision-language contrast: single-view *vs.* multi-view. "single" and "multiple" denote the number of image views used in vision-language contrastive learning.

**Qualitative Analysis.** Figure 2 shows some visualization results of multi-view semantic segmentation consistency for ViewCo and GroupViT. As shown in Figure 2(a), in GroupViT, the semantic segmentations of different views from the same image are inconsistent. For exam-

---

[1]The latest version of GroupViT reports the results of training on the CC3M+CC12M+YFCC dataset.

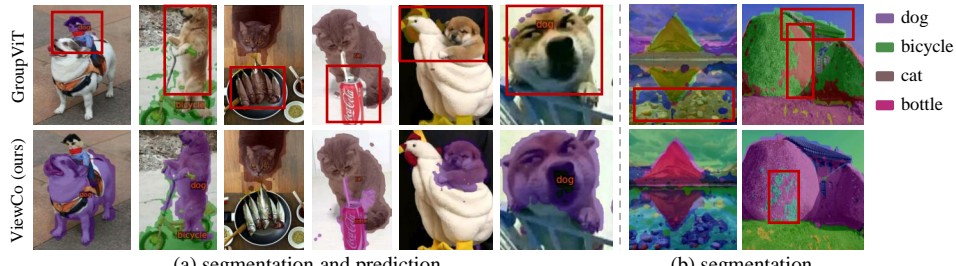

(a) segmentation and prediction         (b) segmentation

Figure 6: Comparison of semantic segmentation of images in rare scenes. (a) Image segmentation and semantic prediction. (b) Image segmentation. ViewCo can better learn high-level cross-modal semantic alignment with the help of two consistency modeling schemes.

ple, in image $x$, "umbrella" is misclassified as "cow", and in view $u$, "umbrella" is misclassified as "horse". There is also the problem of inconsistent semantic segmentations between views $u$ and $v$. As shown in Figure 2(b), the semantic segmentation in different views in ViewCo is completely consistent. This shows that our cross-view segmentation consistency modeling and text-to-views consistency modeling in ViewCo are effective.

To evaluate ViewCo's ability to perform semantic segmentation through semantic understanding in rare scenes, we show the more visual comparison in Figure 6. The images of rare scenes are selected from the Internet. In Figure 6(a), we use the class labels of the PASCAL VOC 2012 dataset as the label set for the images. ViewCo's segmentation and prediction results in rare scenes are significantly better than

| | Pre-training dataset | Zero-shot Acc@1 (%) | Acc@5 (%) |
|---|---|---|---|
| GroupViT | CC12M | 37.5 | 65.5 |
| ViewCo | CC12M | **39.5** (2.0↑) | **68.4** (2.9↑) |
| ViT | CC12M+YFCC | 42.4 | - |
| GroupViT | CC12M+YFCC | 42.9 | 71.7 |
| ViewCo | CC12M+YFCC | **46.3** (3.4↑) | **74.0** (2.3 ↑) |

Table 5: Zero-shot performance on ImageNet.

GroupViT's. This indicates that ViewCo can better understand high-level semantics in images through consistent semantic learning. In Figure 6(b), we only focus on the model's ability to segment images in rare scenes. Compared to GroupViT, ViewCo handles the details of image segmentation much better.

More visual comparison results are shown in Figure 7 of A.2 of the supplementary material. In addition, we also visually compare the segmentation consistency of ViewCo and GroupViT on different views in A.3. Finally, we present an analysis of ViewCo's cross-view segmentation consistency in A.4.

**Image Classification.** We also evaluate the classification performance of ViewCo. As shown in Table 5, ViewCo significantly outperforms ViT (*i.e.*, 46.3% *vs.* 42.4%) and GroupViT (*i.e.*, 46.3% *vs.* 42.9%), showing that ViewCo achieves better cross-modal semantic alignment through text-to-views consistency modeling.

## 5   CONCLUSION

We propose a novel and simple multi-view consistency learning (ViewCo) for text-supervised semantic segmentation. To deal with the problems of excessively strict image-text correspondence and ambiguous text supervision in the VLP model, ViewCo models the text-to-views consistency and cross-view segmentation consistency. ViewCo can generate consistent segmentations and better capture high-level cross-modal semantic alignment. We expect that this exploration of multi-view consistent learning is also applicable to other VLP tasks.

## 6   ACKNOWLEDGMENT

This work was supported in part by National Key R&D Program of China under Grant No.2020AAA0109700, National Natural Science Foundation of China (NSFC) under Grant No.61976233, Guangdong Outstanding Youth Fund (Grant No.2021B1515020061), Shenzhen Fundamental Research Program (Project No.RCYX20200714114642083, No.JCYJ20190807154211365). We thank MindSpore and CAAI-Huawei MindSpore Open Fund for the partial support of this work, which is a new deep learning computing framwork[2].

---

[2]https://www.mindspore.cn

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
