# OpenReview forum: "ViewCo: Discovering Text-Supervised Segmentation Masks via Multi-View Semantic Consistency"
_ICLR.cc/2023/Conference — ICLR 2023 poster_

### Official Review · Reviewer_kMrx · 2022-10-25

**Confidence:** 3
**Correctness:** 4
**Technical Novelty And Significance:** 4
**Empirical Novelty And Significance:** 4
**Recommendation:** 8

**Clarity, Quality, Novelty And Reproducibility:**

Quality: The paper has a good quality and is evaluated extensively.

Clarity: The paper is written well and easy to understand.

Originality: The proposed method is new.


**Strength And Weaknesses:**

Strength

-The idea of multi-view consistency is new in visual-language learning. This paper proposes a new method to implement the idea.

-This paper outperforms GroupVIT consistently on multiple benchmarks.

Weakness

-A potential related work is missing. The DenseCL [A] paper studies multi-view consistency for self-supervised contrastive learning. The author should discuss the difference.

[A] Dense Contrastive Learning for Self-Supervised Visual Pre-Training. Want et al., CVPR'21

**Summary Of The Paper:**

This paper proposes to improve text-supervised segmentation model i.e., GroupVIT by enforcing multi-view semantic consistency. Specifically, a text-to-views consistency module is included to encourage different views to be aligned with the same text input. In addition, there is a cross-view consistency module which enforces features from the different views of the same input image to be closer. The authors conduct extensive experiments on the semantic segmentation benchmarks and show SOTA results.

**Summary Of The Review:**

Overall, I think this is a good paper with sufficient novelty and good results. I would recommend to accept it.

---

> ### Author Response · Authors · 2022-11-18
> **To Reviewer kMrx:**
>
> We are grateful for your comprehensive and encouraging review!
>
> **R4Q1.	A comparison with related work DenseCL is missing.**
>
> Thank you for these valuable comments. The self-supervision learning method of DenseCL is based on pixel-level contrastive learning. In contrast, ViewCo uses higher-level local semantics as the self-supervised information of the image to provide a more stable feature representation for the model, and at the same time help ViewCo obtain better zero-shot transfer capabilities.
> We have added an analysis of the related work DenseCL in Section 2 Consistent Semantics of the modified version.

---

### Official Review · Reviewer_wv76 · 2022-10-26

**Confidence:** 4
**Correctness:** 3
**Technical Novelty And Significance:** 3
**Empirical Novelty And Significance:** 3
**Recommendation:** 8

**Clarity, Quality, Novelty And Reproducibility:**

There are many hyperparameters and implemental details in the new system, so it could be difficult to reproduce the experimental results without the author's source code.

**Strength And Weaknesses:**

Strength
+ Compared to the baselines and other methods, the proposed methods could bring some improvements in accuracy.
+ Some ablation experiments are introduced to prove the effectiveness of the proposed method.

Weaknesses
-  The definition of positive pairs and negative pairs in the cross-view consistency module is missing.


**Summary Of The Paper:**

This paper studies text-supervised semantic segmentation. The authors propose multi-View Consistent learning (ViewCo) to introduce the correspondence among multiple augmented views of the same image. The experimental results on serval datasets demonstrate the effectiveness of the proposed method.

**Summary Of The Review:**

Overall, it is a solid paper and gives a reasonable idea to improve text-supervised semantic segmentation.

---

> ### Author Response · Authors · 2022-11-18
> **To Reviewer wv76:**
>
> We are grateful for your comprehensive and encouraging review!
>
> **R3Q1. The definition of positive pairs and negative pairs in the cross-view consistency module is missing.**
>
> Thank you for this comment and sorry for the confusion. We would like to point out that the definitions of the positive and negative pairs are given in Section 3.1 below Eq. (2). In addition, for clarity, in this paragraph of the revised version, we have also added the reason why the segment tokens on the corresponding position IDs of different views have the same semantics.

---

### Official Review · Reviewer_cqK1 · 2022-11-02

**Confidence:** 4
**Correctness:** 2
**Technical Novelty And Significance:** 2
**Empirical Novelty And Significance:** 2
**Recommendation:** 3

**Clarity, Quality, Novelty And Reproducibility:**

The paper is well-written, and the motivation is clear. However, the novelty is limited, and the major assumption doesn't hold.

**Strength And Weaknesses:**

### Strength:

(1) This work improves Group VIT by enforcing view consistency

(2) Author conducts substantial experiments across multiple datasets and shows consistent quantitative improvement over the baseline.

### Weaknesses:

(1)  I think the assumption of cross-view segment consistency by segment id is wrong. As observed in other VIT frameworks, which use learnable tokens for instance segmentation or objective detection, the tokens usually have spatial rather than semantic biases. Based on that, the segment id primarily indicates the segmentation's relative spatial location to the image, and matching group ids does not make sense (we can simply horizontally and vertically flip the images to break the assignment).

Instead of directly matching segment ids, I think reasonable alternatives are: (a) Compute bipartite matching based on group features similarity and (b) Spatially warp one view to another and compute the segmentation overlapping to match the segments. Then ignore the segment with zero overlapped pixels.

**Summary Of The Paper:**

This paper improves Group VIT, a text-supervised segmentation model, by introducing region-wise cross-view consistent regularization. The author benchmarked for zero-shot segmentation capability across multiple datasets and showed improvements over several baselines.

My primary concern with this work is selecting the positive segmentation across views, where the author assumes the segments with the same id form positive pairs. I think this assumption is wrong, and I'll expand the discussion to the Weaknesses section.

This work's contribution is also limited since most of its success is built upon the Group VIT model. The quantitative improvements from the newly introduced regularization are not substantial, around 1-2% over Group VIT.







**Summary Of The Review:**

Given my concerns over the assumption of the objectives and the limited improvement over the baseline, I tend to reject this submission.

I'll reconsider my rate if the author can justify why using segment id for cross-view consistency makes sense or provide experimental results to show that segment ids indicate semantic categories rather than spatial bases.

---

> ### Author Response · Authors · 2022-11-18
> **To Reviewer cqK1:**
>
> Thank you for your detailed comments. We explain the rationality of our cross-view segment consistency in detail below. We sincerely hope you could reconsider the significance of our work.
>
> **R2Q1. The rationality of cross-views segmentation consistency module.**
>
> Thank you for this valuable comment and sorry for the confusion. The semantic consistency  of the segment tokens with the same segment IDs is explained in the following two aspects.
>
> First, the learnable tokens used by ViewCo are position-independent when grouping image pixels, and augmented operations such as flipping do not affect the semantic extraction of images. Second, unlike previous fully supervised ViT frameworks [1] based on learnable tokens, ViewCo's cross-view segmentation consistency module uses a self-supervised mechanism based on the EMA policy for parameter updates.  These learnable group tokens are used as the query of the Grouping Block attention module and are also a kind of model parameter, so the semantics of segment tokens obtained on the corresponding position IDs of different views are highly related. In addition, due to the translation invariance and flipping invariance of the attention mechanism, the output segment tokens are also translation invariant and flipping invariant. That is to say, flipping does not affect the order of the segment tokens and thus does not affect the correspondence of the segment tokens with the same IDs.
> This ensures the rationality of the cross-view segmentation consistency module design, which is also verified by Figure 2 and Table 5 in the paper. We have added a clear explanation in the paragraph under Eq. (2) in  Section 3.1 of the modified version. In addition, we have added a visual comparison between ViewCo and GroupViT on segmentation consistency of multiple views in Figure 8 of Appendix A.2 in the supplementary material.
>
>
> In addition, we also tried to use the segmentation consistency of the intersecting regions of the two views as self-supervision information, but unfortunately, we were not successful because the pixels in the intersecting regions are predicted to be the same semantically yielding trivial solutions. Similar “shortcut" phenomena are also observed by [2,3]. In the future, we will explore different matching mechanisms.
>
> [1]Carion, Nicolas, et al. “End-to-end object detection with transformers." European conference on computer vision. Springer, Cham, 2020.
>
> [2]Chen, Xinlei, and Kaiming He. “Exploring simple siamese representation learning." Proceedings of the IEEE/CVF Conference on Computer Vision and Pattern Recognition. 2021.
>
> [3]Grill, Jean-Bastien, et al. “Bootstrap your own latent-a new approach to self-supervised learning." Advances in neural information processing systems 33 (2020): 21271-21284.

---

> > ### Comment · Reviewer_cqK1 · 2022-11-25
> > **Thanks for the comments but your reply doesn't resolve my concerns.**
> >
> > Thanks for the author's comment and for providing the visualization.
> >
> > However, I don't think the author's justification resolves my concerns on the rationality of cross-view matching using token-id, and I’m still worried that the major assumption of this paper is wrong.
> >
> > > Learnable tokens are position-independent
> >
> > I don't think this claim is correct without further justification or verification via the experimental result. The learnable token in another VIT framework [1] is clearly position-dependent. [1] uses bi-partite matching rather than manually assigned targets. Therefore, VIT has the freedom to make their object-token either semantic-biased or positional-biased. From their visualization, the token is learned to have positional bias, and I'm worried that the same thing happened in ViewCo.
> >
> >  > ViewCo uses EMA to update model's parameters as well as the learnable token
> >
> > I don’t think the updating scheme is the key to shifting the learnable token from learning positional-biased to semantical-biased features. As in the reply above, the group token can be positional-biased via gradient descent.
> >
> > > Attention mechanism is invariant to flipping and translation so does the segment token.
> >
> > I think this claim is wrong. In VIT, the output will change unless you flip or translate the positional encoding accordingly. Another concern is what if one object in one view is missed in another view? In such cases, the view-consistency assumption doesn't make sense regardless of the matching strategy.
> >
> >
> > > Visual comparison to GroupVIT in the supplementary material.
> >
> > Thanks for providing the visualization. I think it's convincing if you can plot the segment id of two augmented views, especially under flipping, and show the ID are invariant to the augmentation.
> >
> >
> > [1]Carion, Nicolas, et al. “End-to-end object detection with transformers." European conference on computer vision. Springer, Cham, 2020.

---

> > > ### Author Response · Authors · 2022-11-26
> > > **Further explanation**
> > >
> > > Thank you for your reply. We provide a detailed explanation and validation of the positional correlation and view consistency assumptions. We hope our response addresses your concerns.
> > >
> > > **The positional correlation of learnable tokens.**
> > >
> > > DETR's bi-partite matching aims to solve the matching problem between the predicted boxes and the ground truth boxes, which are strongly associated with the position. Different from DETR, the grouping block in ViewCo groups patch tokens in the way of gumbel softmax through learnable group tokens. The patch tokens of the same group are added element-wise and linearly projected to obtain the corresponding segment tokens. Therefore, the semantics of segment tokens are only associated with learnable group tokens and are independent of the original position of patch tokens in the image. This ensures that positional-biased does not have an impact on learnable tokens. In addition, the EMA parameter update policy ensures the consistency of learnable tokens on the corresponding position IDs of different views, so the semantics of segment tokens corresponding to position IDs are consistent. Finally, based on your suggestion, we visualized the semantic consistency of segment IDs in different views (especially flipped views), which is shown in this link https://radial-fire-e77.notion.site/ViewCo-84e25a2d2d1b428bbe2545cda08ce0af.
> > >
> > > **Rationality of view consistency assumption.**
> > >
> > > The semantic consistency of multi-crop has been shown to be effective in many image self-supervised works[1][2]. In ViewCo, we try to extend this semantic consistency to the segment level. It is foreseeable that ViewCo does not guarantee that the two views will have an intersecting area. In fact, the design motivation of ViewCo's cross-view segmentation consistency module is not based on the assumption of view intersection. Instead, we expect ViewCo to capture and align the same semantics from disjoint multiple views, forcing the model to capture higher-level semantics by accomplishing this challenging task. This is reasonable because just as humans can identify object categories based on local regions of objects, we also expect models to recognize and align these semantics based on local features of objects from two disjoint views. Our experiments also demonstrate the plausibility of this mechanism. For example, the convergence speed and performance of ViewCo trained on GCC12M is much higher than that of GroupViT under the same settings (e.g., when epoch=16, the mIoU of the two methods on the VOC dataset is 45.5% and 36.0%, respectively). At the same time, by replacing the semantic-level alignment with the view-level, ViewCo's mIoU performance on VOC drops by 1.5%. The results in Table 4 of the ablation experiments also show this.
> > >
> > > [1] Zhou, Jinghao, et al. "Image BERT Pre-training with Online Tokenizer." *International Conference on Learning Representations*. 2021.
> > >
> > > [2]Caron, Mathilde, et al. "Emerging properties in self-supervised vision transformers." *Proceedings of the IEEE/CVF International Conference on Computer Vision*. 2021.

---

> > > > ### Comment · Reviewer_cqK1 · 2022-12-05
> > > > **Reply to Author**
> > > >
> > > > > DETR's bipartite introduces spatial correlation while ViewCo doesn’t
> > > >
> > > > I disagree with this claim. In DETR, the order of output tokens can be arbitrarily permutated, e.g., the prediction could be ordered by hyper semantic categories, the case author prefers, or by location, the case I referred to in my previous reply. In both cases, bipartite matching would return the same result regardless of the prediction order, and therefore the matching objectives won't introduce spatial biases. The best explanation for DETR is that the box tokens learn spatial biases, which help recognize the boundary of the instance. In ViewCo, the grouping operations are not fundamentally different from DETR's internal token-patch interaction. If that is the case, the Gumbel softmax operation will cooperate with token information and adjust accordingly to include spatial information, which is helpful for the general recognition task.
> > > >
> > > > Regardless of whether the tokens are spatially biased, I’m concerned about the author's reply, "This ensures that positional-biased does not have an impact on learnable tokens." I don't think grouping without spatial information is a correct design choice, and motivating this submission from this perspective would deepen my concerns.
> > > >
> > > > Thanks for providing additional visualization on augmented samples, but those are not convincing for the following reasons: (1) The visualization only has four post-processed tokens rather than the total eight tokens of the second stage. (2) The image is tricky since the two sheep's positions are horizontally and vertically symmetrical. It's better to show visualization on the images without symmetrical bias. Understandably, the current method can't get decent token-wise segmentation on the challenging data, but visualization of those cases can be helpful for readers to understand the internal mechanism.
> > > >
> > > > I agree with reviewer Dedj that introducing a view-consistency target to the existing system raises concerns about novelty. At the same time, the authors don't seem to provide a convincing analysis for their design choice of segment-aware consistency target. Overall, I think this paper needs substantial improvement in its methodology and analysis to meet ICLR standards. Hence, I maintain my recommendation of rejection.

---

> > > > > ### Author Response · Authors · 2022-12-07
> > > > > **Focus on visualization**
> > > > >
> > > > > Dear reviewers,
> > > > >
> > > > > Thank you very much for your patience and prompt reply.
> > > > >
> > > > > I understand that your concern mainly comes from whether the segments on the same location IDs output by the teacher and student networks from different views of the same image have the same semantics. We try to prove this through experiments, and the latest visualization results are shown in the following link [https://radial-fire-e77.notion.site/ViewCo-84e25a2d2d1b428bbe2545cda08ce0af](https://www.notion.so/ViewCo-84e25a2d2d1b428bbe2545cda08ce0af). As shown in Figure 1, we avoid those images with symmetric semantics to avoid unnecessary interference to the visualization results. Specifically, we visualized the segmentation results of the 8 segment semantics output by the second grouping block in the teacher and student networks of different views. It should be noted that we did not perform any post-processing on the visualization results, but only numbered the main semantics in the image for the convenience of visualization. Image segmentation shows less than 8 semantics, mainly because in the grouping block the model determines which segment semantics each image patch belongs to by computing an attention map of shape 196 x 8 (196 is the number of image patches). This means that some segments may not have corresponding image patches. Second, in the segmentation results from different views of the same image, image regions with the same color represent the same location IDs and the same semantics. From the visualization results in Figure 1, it can be found that the semantics in the segmentation maps of different views output by the teacher and student networks maintain a good consistency in the corresponding position IDs, which shows that the design of ViewCo's cross-view segmentation consistency module is reasonable.
> > > > >
> > > > > We thank you again for your patience and look forward to hearing from you.

---

> > > > > > ### Comment · Reviewer_cqK1 · 2022-12-10
> > > > > > **Reply to Author**
> > > > > >
> > > > > > I highly appreciate the author's efforts in providing more visualization to showcase the ID and region association. The token IDs are organized implicitly in order to maximize the ID-wise learning objectives. This is an interesting observation, and I recommend including those figures in the paper.
> > > > > >
> > > > > > However, it only works when two augmented views contain exactly the same semantic categories; otherwise, any missing or extra semantic category in one view will trigger the IDs shifting and break the alignment. In this case, I think extra postprocessing is necessary to robustify the alignment, especially for scene images.
> > > > > >
> > > > > > Overall, the author's reply answers some of my concerns, but extra improvements over the existing method are needed to achieve claimed region-wise alignment and showcase the substantial contribution on top of the baseline methods. Therefore, I think the current version is not ready for publishing in ICLR, and I tend to reject this submission.

---

> > > > > ### Author Response · Authors · 2022-12-10
> > > > > **Looking forward to hearing from you**
> > > > >
> > > > > Deer reviewers,
> > > > >
> > > > > Thank you very much for your patience and prompt reply.
> > > > >
> > > > > We have updated the results with more adequate visualizations, which we hope will address your main concern about segmentation consistency across views. We look forward to receiving your response.

---

### Official Review · Reviewer_Dedj · 2022-11-09

**Confidence:** 4
**Correctness:** 2
**Technical Novelty And Significance:** 2
**Empirical Novelty And Significance:** 2
**Recommendation:** 3

**Clarity, Quality, Novelty And Reproducibility:**

The paper is easy to follow. The novelty is limited due to the concerns raised in the Strength And Weaknesses section. The code is not provided for reproducibility.

**Strength And Weaknesses:**


Strength:

+ The motivation of introducing multi-view consistency constrain to text-supervised semantic segmentation is reasonable.

+ Experimental reuslts on multiple datasets show some benifits compared with existing methods.


Concerns:

- The general idea of introducing multi-view consistency constrains to vision-language pretraining models is not new. Some existing works, e.g. DeCLIP [a] have explored adding such multi-view self-supervision constrains and validated its effectiveness when learning vision-language models. The differences here are mainly about applying multi-view constrains on segment level rather than whole image when building upon the framework of Group VIT.

[a] Supervision exists everywhere: A data efficient contrastive language-image pre-training paradigm, ICLR 2022

-  Cross-view constrains are applied on segments of two augmented views. Since there is no guarantee for location (segments) correspondence between these two augmented views, the proposed strategy in Figure 5 of constructing positive and negative pairs for the corss-view segments seems not reasonable.

- The performance of the most important baseline GroupViT seems not consistent with the original paper. For example, in Table 1 Comparison with zero-shot baselines on PASCAL VOC 2012, it reports 51.2 mIoU for GroupViT and 52.4 mIoU for the proposed method ViewCo, while in the original paper [b] GroupViT obtains 52.3 mIoU, which is much higher than 51.2 reported in this submission and is only 0.1 lower than the proposed method. Besdies, on PASCAL Context, GroupViT reports 22.4 mIoU, the improvements from ViewCo seems also marginal (23.0 mIoU v.s. 22.4 mIoU).

[b] GroupViT: Semantic Segmentation Emerges from Text Supervision, CVPR 2022

**Summary Of The Paper:**

This paper focuses on text-supervised semantic segmentation by studying a multi-view consistency learning framework. Specifically, a text-to-views consistency and a cross-view segmentation consistency training strategy are proposed and added upon existing work Group VIT. The proposed method tries to construct more training constrains based on correspondence among multiple augmented views to obtain more improvements. Experiments are conducted on multiple datasets for zero-shot segmentation, e.g. PASCAL and COCO., showing improvements upon some existing baselines.


**Summary Of The Review:**

My current rating is based on concerns for limited technical contributions, unreasonable positive/negative pair selection for cross views, quite limited improvments upon baseline GroupViT. Please find more details in the Strength And Weaknesses section.

---

> ### Author Response · Authors · 2022-11-18
> **To Reviewer Dedj:**
>
> Thanks for your detailed and constructive comments. We respond to all the issues you pointed out in detail below. We hope our response and rebuttal revision will address your concerns.
>
> **R1Q1. ViewCo vs. DeCLIP.**
>
> Thank you for this valuable question and sorry for the confusion. We would like to point out the technical difference between our ViewCo and DeCLIP in the following aspects: First, on a technical basis, DeCLIP believes that the text may only describe a part of the image, so an additional local view is introduced for the image branch. ViewCo mainly hopes to mine consistent semantics from images for segmentation through the one-to-many alignment of cross-modal text-to-views. Secondly, in terms of technical implementation, they have the following three obvious differences that lead to completely different model preferences:
>
> 1. For image augmentation, DeCLIP uses a global view and a local view for self-supervised learning, but this causes information leakage to reduce the quality of features and affects the alignment of cross-modal semantics. In contrast, ViewCo takes the segmentation consistency of two local views as self-supervised information, which not only ensures the difficulty of the task but also helps the model to find the corresponding implicit consistent semantics for segmentation.
> 2. In the self-supervision of the image branch, compared with DeCLIP using SimSiam as the supervision mechanism, we use EMA to ensure the semantic consistency of the segments on the corresponding position IDs of different views, while helping the model avoid distractions from ambiguous text annotations.
> 3. For text augmentation, DeCLIP uses EDA [1] as a text augmentation strategy. The augmented text still contains multiple semantics, which is not conducive to the alignment of local semantics in segmentation tasks. In contrast, ViewCo uses two local views for self-supervision, while using a “prompt engineering" mechanism to obtain an augmented text with a single semantic. Combining one-to-many alignment can help ViewCo to better mine consistent segmentation semantics in images.
>
> Given the above observations, we believe that DeCLIP and ViewCo are fundamentally different in both motivation and implementation. In addition, we have added a detailed comparison between ViewCo and DeCLIP at the end of Section 3.2 of the revised version.
>
> [1] Wei J, Zou K. EDA: Easy Data Augmentation Techniques for Boosting Performance on Text Classification Tasks[C] EMNLP-IJCNLP. 2019: 6382-6388.
>
> **R1Q2. The rationality of cross-views segmentation consistency module.**
>
> Thank you for this valuable comment. Unlike previous fully supervised ViT frameworks [1] based on learnable tokens, ViewCo's cross-view segmentation consistency module uses a self-supervised mechanism based on the EMA policy for parameter updates.
> These learnable group tokens are used as the query of the Grouping Block attention module and are also a kind of model parameter, so the semantics of segment tokens obtained on the corresponding position IDs of different views are highly related.
> In addition, due to the translation invariance and flipping invariance of the attention mechanism, the output segment tokens are also translation invariant and flipping invariant. That is to say, flipping does not affect the order of the segment tokens and thus does not affect the correspondence of the segment tokens with the same IDs. This ensures the rationality of the cross-view segmentation consistency module design, which is also verified by Figure 2 and Table 5 in the paper. We have added a clear explanation in the paragraph under Eq. (2) in  Section 3.1 of the modified version. In addition, we have added a visual comparison between ViewCo and GroupViT on segmentation consistency of multiple views in Figure 8 of Appendix A.2 in the supplementary material. Further, we visualized the semantic consistency of segment IDs in different views (especially flipped views), which is shown in this link https://radial-fire-e77.notion.site/ViewCo-84e25a2d2d1b428bbe2545cda08ce0af.
>
> [1] Carion, Nicolas, et al. “End-to-end object detection with transformers." European Conference on Computer Vision. Springer, Cham, 2020.
>
> **R1Q3. The performance of GroupViT differs from the original text.**
>
> Thank you for this comment and sorry for the confusion. We would like to clarify that the results of the updated version of the GroupViT paper are all trained on CC3M + CC12M + YFCC14M, which is different from our settings. We report the performance of ViewCo and GroupViT trained on CC12M+YFCC14M and zero-shot transfer to VOC (52.4 *vs.* 51.2, the results of GroupViT can be found at this link https://arxiv.org/pdf/2202.11094v1.pdf). We have added a footnote at the bottom of the page with Table 2 in the revision to clarify the difference caused by the update of the GroupViT paper.

---

> > ### Comment · Reviewer_Dedj · 2022-12-09
> > **Thanks for authors' reply**
> >
> > I appreciate response from authors. My major concern remains after reading authors’ responses as well as other reviews.
> >
> > Regarding the technical contributions, the general design is shared with existing work as author listed here. There are 1-1 correspondences for image augmentation, self-supervision of the image branch, text augmentation. The specific implementation details or choice of augmentation strategy are not well justified as new technical contributions, such as using EMA or SimSiam, using EDA as text augmentation strategy or “prompt engineering” , etc. The claimed strengths of such choices are also not well supported by experiments. Introduceing cross view consistency constrain is also upon an existing system.
> >
> > Regarding the performance comparisons between ViewCo and the important baseline GroupViT.  GroupViT’s 52.3 performance can be found since the version of May 19 2022 https://arxiv.org/pdf/2202.11094v2.pdf, even much earlier than the CVPR proceeding. It is not convincing to closely follow GroupViT as baseline while not follow its experimental setting in the camera ready version (released in May).
> >
> > To summarize, I do not think this paper is in a good shape to be accepted by ICLR, therefore, I keep my suggestion of rejection.

---

> > > ### Author Response · Authors · 2022-12-09
> > > **Further clarification**
> > >
> > > Dear reviewers,
> > >
> > > Thank you very much for your prompt reply.
> > >
> > > For the setting of GroupViT experimental results, it can be found in GroupViT's official source code setting [https://github.com/NVlabs/GroupViT/blob/13b786155a1dfffe4703f40d028c92be58e1178d/configs/default.yml#L36](https://github.com/NVlabs/GroupViT/blob/13b786155a1dfffe4703f40d028c92be58e1178d/configs/default.yml#L36) that 52.3% of the experimental results on VOC are based on the results on the GCC3M+GCC12M+YFCC14M dataset.
> > >
> > > It should be pointed out that, similar to CLIP, DeCLIP focuses on visual language pre-training and mainly evaluates model performance on classification tasks. In contrast, ViewCo's model design and technical details are mainly aimed at zero-shot semantic segmentation tasks, so GroupViT is used as the main baseline. In addition, since DeCLIP uses a stronger image encoder (i.e., ViT-B, while ViewCo and GroupViT use ViT-T), their results cannot be directly compared. And their technical advantages in their respective tasks do not conflict.
> > >
> > > Thank you again for your prompt reply, and we hope the above statement addresses your concerns in a timely manner.

---

### Author Response · Authors · 2022-11-18
**General Response to All Reviewers**

We thank all the reviewers for their time, insightful suggestions, and valuable comments. We are grateful for the positive recognition of the reviewers on our motivation (Dedj), novelty (kMrx), extensive experiments (cqK1 and wv76), and good results (Dedj, wv76, and kMrx). Reviewers Dedj and cqK1's concerns may be based on some incomplete observations. We respond to each reviewer's comments in detail below. We have also revised the main paper and the supplementary material according to the reviewers' suggestions. The main changes are listed as follows:

- ViewCo *vs.* DeCLIP. We have compared and analyzed the technical implementation of DeCLIP and ViewCo in detail from three aspects: image augmentation, self-supervised strategy, and text augmentation, and there are essential differences between them. At the end of Section 3.2, we add a detailed comparison of ViewCo with DeCLIP (to Reviewer Dedj).

- Semantic consistency of segments on the corresponding position IDs. ViewCo updates the parameters of the image branch based on the EMA strategy. This makes the learnable group tokens on the corresponding position IDs in different views highly correlated, thus ensuring the semantic consistency of the segments on the corresponding position IDs. In the paragraph under Eq. (2) in Section 3.1, we clarify the reason why the segment tokens on the corresponding position IDs of different views have the same semantics. Furthermore, we add a visual comparison between ViewCo and GroupViT on segmentation consistency of multiple views in Figure 8 of Appendix A.2. (to Reviewers Dedj, cqK1, and wv76).
- Inconsistent reporting performance. The updated version of GroupViT reports the results on CC3M+CC12M+YFCC, which is different from our setting. Under the same setting, ViewCo trained on CC12M+YFCC and zero-shot transferred to VOC is significantly better than GroupViT (52.4 *vs.* 51.2). At the bottom of the page with Table 2, we add a footnote to clarify that this performance difference is due to different training set settings (to Reviewer Dedj).
- DenseCL is based on dense contrastive learning at the pixel level, focusing more on local information. In contrast, ViewCo is based on segment-based high-level semantic contrastive learning, which helps to improve the zero-shot transfer capability of the model. In Consistent Semantics in Section 2, we add an analysis of DenseCL (to Reviewer kMrx).
- We again have checked and revised the expression of the full text.

Note that we have marked all the changes in blue. We hope our response and rebuttal revision will address the reviewers' concerns.

---

### Decision · Program_Chairs · 2023-01-20

**Decision:**

Accept: poster

**Justification For Why Not Higher Score:**

The paper, and in particular the initial submission, has done shortcomings indicated in the reviews. Some of these have been addressed in the modified paper, but the final revision is yet to be made, so I would be a bit conservative about promoting the paper. That said, I probably wouldn't mind if this were a spotlight.

**Justification For Why Not Lower Score:**

Enough novelty in the main idea to justify acceptance.

**Metareview: Summary, Strengths And Weaknesses:**

I think the paper is above car for acceptance. TTIC is despite two of the reviewers giving it very low scores. In the ensuing discussion there was a lot of new information and clarifications provided by the authors, and the modifications on the paper have significantly improved it. I think the key idea of using multi view consistency in these settings is sufficient novel, and the results convincing enough to make this a worthwhile part of the conference. I very strongly encourage the authors to take all the comments by the expert reviewers to heart, and to incorporate as much material from the discussion into the revised paper as possible.

**Note From Pc:**

if the above contains the word "oral" or "spotlight" please see: "oral" presentation means -> notable-top-5% and "spotlight" means -> notable-top-25%. As stated in our emails, we are disassociating presentation type from AC recommendations